# Evaluation of Amisulbrom Products for the Management of Clubroot of Canola (*Brassica napus*)

**DOI:** 10.3390/plants13010028

**Published:** 2023-12-21

**Authors:** Zhiyu Yu, Stephen E. Strelkov, Sheau-Fang Hwang

**Affiliations:** Department of Agricultural, Food and Nutritional Science, University of Alberta, Edmonton, AB T6G 2P5, Canada; zhiyu3@ualberta.ca (Z.Y.); strelkov@ualberta.ca (S.E.S.)

**Keywords:** *Brassica napus*, clubroot, disease management, fungicides, *Plasmodiophora brassicae*, QiI

## Abstract

Clubroot, caused by *Plasmodiophora brassicae*, is an important disease of canola (*Brassica napus*). Amisulbrom, a quinone inside inhibitor (QiI), was evaluated for its effectiveness in clubroot management in Alberta, Canada. Resting spores of *P. brassicae* were treated in vitro with 0, 0.01, 0.1, 1, and 10% (*w*/*v*) amisulbrom to determine its effect on spore germination and viability. Amisulbrom inhibited resting spore germination by up to 79% and reduced viable spores by 31% relative to the control. Applications of a liquid solution (AL1000, 1000 g active ingredient (ai) ha^−1^) and granular formulations (AF700, 700 g ai ha^−1^; AF1000, 1000 g ai ha^−1^; AF1500, 1500 g ai ha^−1^) of amisulbrom were tested on the canola cultivars ‘45H31’ (clubroot-susceptible) and ‘CS2000’ (moderately resistant) under greenhouse conditions and in field experiments in 2019 and 2020. In the greenhouse, the treatments were evaluated at inoculum concentrations of 1 × 10^5^ or 1 × 10^7^ resting spores g^−1^ soil. A trend of decreasing clubroot severity with an increasing amisulbrom rate was observed. At the lower spore concentration, treatment with AF1500 resulted in a clubroot disease severity index (DSI) <20% for both cultivars, while the lowest DSI under both low and high spore concentrations was obtained with AL1000. The field results indicated a significant reduction in DSI, with varied effects of rates and liquid vs. granular formulations. The greatest reductions (up to 58.3%) in DSI were obtained with AF1500 and AL1000 in 2020. These findings suggest that amisulbrom holds promise as part of an integrated clubroot management approach.

## 1. Introduction

Canola (oilseed rape, *Brassica napus* L.) is an important crop, accounting for approximately 12% of the total oilseed production worldwide [1]. While Canada is a major producer of canola, in recent years, the emergence of clubroot disease, caused by *Plasmodiophora brassicae* Woronin, has posed a significant threat to yields [2]. The disease is characterized by the development of galls or clubs on the roots of infected plants, which can severely limit growth and yield. The pathogen can produce large numbers of resting spores in infected host tissues, which are released back into the soil as the galls decay, persisting for many years and serving as inoculum for future infections [3,4].

Sustainable clubroot management requires an integrated approach [2,5]. Currently, clubroot control in western Canada relies heavily on resistant cultivars. Unfortunately, the emergence of resistance-breaking pathotypes of *P. brassicae* suggests that additional strategies are needed to complement host resistance [6,7,8]. The soil fumigant metam sodium (Vapam) has proven to be an effective chemical for clubroot control [9,10,11]. Multiple jurisdictions, however, have banned this product due to public health concerns [5]; moreover, its cost makes the application of metam sodium feasible only for eradicating isolated infection foci in canola cropping systems [10,11]. Fungicides represent another potential tool for the management of clubroot. Products such as pentachloronitrobenzene (PCNB), trichlamide, flusulfamide, fluazinam, and cyazofamid have been reported to show efficacy against clubroot [5,12,13,14]. 

Mitochondrial respiration inhibitors are preferred for clubroot control, as they have a shorter half-life and act on specific molecular sites of certain orders of microorganisms [5,15]. Protectant fungicides that act on the mitochondrial complex III block electron flow at either of two ubiquinone-binding sites (Qo—quinone outside and Qi—quinone inside) and are therefore classified as Qo and Qi inhibitors (QoI and QiI) [15]. Flusulfamide (QoI) and cyazofamid (QiI) have been reported to inhibit *P. brassicae* resting spore germination [13,14], and field experiments with these products in Canada indicated some efficacy in reducing clubroot severity on canola [16,17]. The QiI cyazofamid is preferred for clubroot management, since it required a lower dosage than the QoI flusulfamide and showed a high specificity on oomycetes and plasmodiophorids [14]. 

Amisulbrom is a relatively novel sulfonamide QiI fungicide used to control various plant diseases, including downy mildews and Phytophthora blights, caused by oomycetes [18,19,20]. While classified in the Rhizaria, *P. brassicae* shares many characteristics as a pathogen with the oomycetes. In oomycetes, amisulbrom restricts the development of zoosporangia and eliminates the mobile zoospores [19,20]. This mode of action suggests that amisulbrom may also be effective against the zoospores and sporangia of *P. brassicae*, thereby helping to prevent both primary and secondary infection. Kawasaki et al. [21] reported that a broadcast application of 0.1% amisulbrom solution at seeding significantly reduced clubroot incidence and severity on Osaka-shirona (*B. rapa* L. ssp. *pekinensis*). Another study by Hollman [8] showed that seed-row applications of an amisulbrom formulation prior to seeding reduced clubroot severity under field and greenhouse conditions. The objectives of this study were to examine the efficacy of granular fertilizer formulations of amisulbrom against clubroot and to compare their effectiveness with a liquid formulation.

## 2. Results

### 2.1. Effect of Amisulbrom on Resting Spore Germination 

A reduction in *P. brassicae* resting spore germination relative to the control was observed in all of the amisulbrom treatments at all of the sampling times (Figure 1a). This reduction was more pronounced as the concentration of amisulbrom increased, with germination rates at 10 days ranging from 63.6% ± 8.5% in the presence of 0.01% (*w*:*v*) amisulbrom to 7.4% ± 2.1% in 10% (*w*:*v*) amisulbrom. When amisulbrom was applied at a rate of 0.1% (*w:v*), which was equivalent to the liquid formulation used in the field and greenhouse trials (1000 g active ingredient (ai) ha^−1^), the percentage of germinated spores after 10 days declined to 34.4% ± 4.0% (SD) compared with 86.7% ± 3.9% in the control treatment. 

### 2.2. Effect of Amisulbrom on Resting Spore Viability 

The inclusion of amisulbrom in the resting spore suspensions reduced their viability at all of the concentrations of the fungicide evaluated (Figure 1b). While spore viability in the untreated control was 70.0% ± 3.8% after 10 days incubation, it was only 42.6% ± 1.7% and 39.1% ± 3.1%, respectively, in the 1% and 10% (*w*:*v*) amisulbrom treatments. At a rate of 0.1% (*w*:*v*) amisulbrom, 51.4% ± 2.3% of the resting spores were viable.

### 2.3. Field Trials

As expected, clubroot severity was lower on the moderately resistant canola ‘CS2000′ compared with the susceptible ‘45H31’ in both 2019 and 2020 (Figure 2). Nonetheless, all treatments with granular or liquid formulations of amisulbrom significantly (*p* < 0.05) reduced clubroot DSI relative to the untreated control (UTC) in both cultivars at all three sites over two years (Figure 2). 

In 2019, the average DSI on the UTCs of ‘45H31’ and ‘CS2000’ was 37.8% and 25.9%, respectively. However, the application of amisulbrom resulted in significant reductions in DSI, with clubroot severity dropping to a range of 6.9% to 13.7% on ‘45H31’ and 5.4% to 9.3% on ‘CS2000’ (Figure 2a). The reductions in DSI obtained with the different rates or formulations of amisulbrom (granular: AF700, 700 g ai ha^−1^; AF1000, 1000 g ai ha^−1^; and liquid: AL1000, 1000 g ai ha^−1^) were not significantly different.

In 2020, clubroot development was more severe than in 2019, with DSIs of 67.1% and 64.5% on the UTC of ‘45H31’ at Sites 1 and 2, respectively, and 37.0% and 38.6% on the ‘CS2000’ controls (Figure 2b,c). As in 2019, the application of amisulbrom significantly reduced DSI relative to the UTCs. However, in addition to significant reductions in DSI between the UTCs and the amisulbrom-treated plots, significant differences were also observed among the different rates and formulations of the fungicide (Figure 2b,c). The greatest reductions in DSI were generally obtained with AF1500 and AL1000. Treatment with AF1500 lowered the DSI on ‘45H31’ to 25.7% and 21.0% at Sites 1 and 2, while AL1000 reduced it to 8.8% and 13.0%, respectively. On ‘CS2000’, treatment with AF1500 reduced the DSI to 4.7% and 8.7% at Sites 1 and 2, respectively, while AL1000 demonstrated a similar effectiveness, reducing it to values ranging from 4.3% to 7.6% (Figure 2b,c). 

Amisulbrom-treated plots exhibited a higher individual plant height, aboveground biomass, and yield compared with the UTCs. In 2019, AF1500-treated plants displayed an approximate 8 cm increase in height and produced 18.2% to 21.2% more aboveground biomass than the UTCs for both cultivars (Table 1 and Table 2). Significant yield improvements were observed only in ‘CS2000’, where the AF700 application resulted in a 50.1% increase in yield relative to the UTC (Table 1 and Table 2). While a numerical increase in yield of up to 52.2% was seen in ‘45H31’, this was not significant. In 2020, plant height in the amisulbrom-treated plots did not differ significantly from the UTCs, except for AL1000 on ‘45H31’ at Site 1 (Table 1 and Table 2). However, AF1500 and AL1000 significantly increased the total biomass for ‘45H31’ at both sites, by up to 89.2% and 96.5%, respectively (Table 1). For ‘CS2000’, these treatments led to a higher numerical biomass at both sites, but significantly increased biomass only at Site 1 (Table 2). Additionally, in 2020, the AF1500 and AL1000 treatments resulted in yield increases of 30.8% to 79.6% for ‘45H31’ and 18.4% to 101.8% for ‘CS2000’, respectively (Table 1 and Table 2).

### 2.4. Greenhouse Trials 

The two runs of the greenhouse experiment showed no significant differences, so the data were pooled for all further analysis. Significant treatment effects were observed for both cultivars at both resting spore concentrations evaluated (Figure 3). At the low spore concentration (1 × 10^5^ resting spores g^−1^ soil mix), all amisulbrom rates or products significantly reduced DSI on the susceptible cultivar ‘45H31’ relative to the UTC. The most substantial reductions, however, were achieved with AF1500 and AL1000, which reduced DSI by 42.6% and 46.5%, respectively (Figure 3a). Similarly, at the high spore concentration (1 × 10^7^ resting spores g^−1^ soil mix), all treatments also resulted in significant reductions in DSI on the susceptible cultivar, with the greatest reduction (63.1% compared with the UTC) obtained with the application of AL1000 (Figure 3a). On the moderately resistant ‘CS2000’, significant reductions in DSI were also observed with all treatments at both inoculum concentrations (Figure 3b). Notably, AL1000 showed the most effective results, with reductions in DSIs of 37.7% and 47.0% on ‘CS2000’ at the lower and higher spore concentrations, respectively (Figure 3b).

Significant differences between the UTC and amisulbrom treatments were also observed with respect to plant height and biomass under greenhouse conditions. At the lower spore concentration, plants of ‘45H31’ treated with AF1000 and AF1500 were 17.1–17.5 cm taller than in the UTC and had a 52.8–62.2% greater biomass (Table 3). Plants of ‘CS2000’ were 11.1–16.6 cm taller in the AF1500 and AF1000 treatments, relative to the UTC, with increases in biomass of 45.8–46.9% (Table 4). At the higher spore concentration, ‘45H31’ plants in the AL1000 treatments were 10.9 cm taller than in the UTC, while AL1000 and AF1500 significantly increased biomass by 33.9% and 43.0%, respectively (Table 3). In the case of ‘CS2000’, treatment with AF1000, AF1500 or AL1000 had similar effects on plant height and biomass; plants were 8.5–9.8 cm taller in these treatments, relative to the UTC, with significant increases in biomass of 26.1–36.2% (Table 4).

### 2.5. Resting Spore Densities 

Amisulbrom treatments did not have a significant effect on *P. brassicae* resting spore density in 2019 nor in 2020 at either field site (Appendix A), although numerical decreases were consistently observed in 2019. The resting spore concentration ranged from 6.6 × 10^4^ to 9.5 × 10^4^ resting spores g^−1^ soil across the field plots in 2019, and from approximately 7.9 × 10^5^ to 8.0 × 10^5^ and 8.0 × 10^5^ to 8.8 × 10^5^ resting spores g^−1^ soil at Sites 1 and 2, respectively, in 2020.

## 3. Discussion

While the management of clubroot on canola relies heavily on the deployment of resistant cultivars, the recent emergence of pathotypes able to overcome this resistance highlights the need for a more integrated disease management approach [7,8,22]. In this study, various amisulbrom treatments were compared for their efficacy in reducing clubroot severity and improving plant growth and yields. In general, amisulbrom showed promise for clubroot control under both field and greenhouse conditions. 

In the field, all of the amisulbrom treatments significantly reduced clubroot severity relative to the UTCs, although no significant differences were observed among the different rates or formulations of the fungicide applied in 2019. In contrast, in 2020, disease severity generally declined further as the rate of amisulbrom increased. This likely reflected more severe clubroot development in 2020 vs. 2019. In 2019, the DSI in the susceptible untreated control remained below 40%, while it exceeded 60% in 2020, allowing for a greater range of disease severities with the different treatments. Ultimately, the DSI in the most effective treatments in both 2019 and 2020 was similar (<10%). It is worth noting that the *P. brassicae* resting spore density was about one order of magnitude greater at the field sites selected in 2020 vs. 2019, which may have contributed to the more severe disease development in the control treatments in 2020 [3,4]. As in the field, all treatments also significantly reduced DSI at the two spore concentrations evaluated in the greenhouse, relative to the UTCs. 

The liquid formulation, AL1000, was one of the most effective treatments in both the field and greenhouse, exhibiting a comparable and sometimes superior efficacy to the granular formulation AF1500, despite the former containing 1000 g ai ha^−1^ vs. the latter’s 1500 g ai ha^−1^. This could reflect a slower release of the active ingredient from the granular form, resulting in lower levels of amisulbrom earlier in the growing season, and therefore reduced protection from early infection by *P. brassicae*; early infection is associated with more severe damage on susceptible hosts [23]. Soil drenches with liquid amisulbrom are frequently used in the production of cruciferous vegetables in small acreage farms in China and Japan [5,24]. However, the in-furrow application of liquid fungicide formulations in the broad-acre canola cropping systems of western Canada may be more challenging, given demands on equipment, time, and labour [25]. In this context, the application of granular amisulbrom is perhaps more practical, since most seeders used by western Canadian growers are capable of applying fertilizer granules into the seed rows when seeding. 

The granular formulations of amisulbrom evaluated in this study included monoammonium phosphate, which was also was applied with the UTC and the liquid formulation at seeding. Monoammonium phosphate releases ammonium (NH_4_^+^) and phosphate (H_2_PO_4_^−^), which could result in a reduction in soil pH [26]. Although there does not appear to be any evidence that this fertilizer or its end-products affect the effectiveness of amisulbrom, a lower soil pH can be more conducive to clubroot development [5]. Kawasaki et al. [21] found that the application of liquid amisulbrom to limed soil provided better clubroot control than its application to soil that had not been treated with lime and almost eliminated disease incidence. Similarly, Nakanishi and Mori [27] also reported that applying a mix of amisulbrom powder and hydrated lime suppressed clubroot development, maintaining the healthy growth of broccoli. These studies, combined with reports of the effectiveness of lime as a clubroot management tool [5,28], indicate that combinations of lime and amisulbrom may hold promise for the improved control of clubroot on canola. 

The consistent reduction in clubroot severity observed on susceptible and moderately resistant canola following amisulbrom treatment under both field and greenhouse conditions suggests the potential of this fungicide to enhance the efficacy and durability of genetic resistance to this disease. Even the most highly resistant canola cultivars may still develop mild symptoms of clubroot, particularly under high inoculum pressure [4,29]. By combining amisulbrom applications with the planting of resistant cultivars, the incidence and severity of clubroot could be diminished further, thereby reducing the amount of inoculum (resting spores) produced in the host and returned to the soil. Since the galls produced in resistant hosts are likely enriched for pathogen genotypes able to overcome that resistance [30], this could slow the emergence of resistance-breaking pathotypes of *P. brassicae*. 

The declines in clubroot severity that were observed with amisulbrom treatment were reflected in an increased plant height and biomass in both the greenhouse and field experiments, and ultimately, in increased yields in the field (as yield was not monitored in the greenhouse). Studies have shown that increased clubroot severity results in a reduction in vegetative growth and yield of canola/oilseed rape [31,32,33]. In Sweden [31] and Germany [32], yield losses in oilseed rape approached 100% under very high clubroot severity. In Canada, yield of canola was reduced between 0.26% and 0.49% for each 1% increment in DSI, regardless of host genetics [33]. Therefore, by reducing clubroot severity, the application of amisulbrom could protect canola yields in clubroot-infested fields.

It has been suggested that amisulbrom and another QiI fungicide, cyazofamid, reduce *P. brassicae* infection by inhibiting the pathogen zoospores and restricting the growth of sporangia [14,18,19,20]. A previous report [34] that used in vitro methodologies similar to this study indicated that *P. brassicae* resting spore germination in a root-exudate solution could exceed 90%, while the average resting spore viability was 84% in sterilized water after 10 days, values consistent with the 87% germination and 70% viability found here in the UTCs. The reductions in resting spore viability and germination with amisulbrom treatments observed in this study indicate that amisulbrom may also reduce clubroot via a direct effect on the resting spores. This hypothesis needs to be further tested, particularly with experiments looking at the effect of amisulbrom on resting spores in the soil, since no significant declines in resting spore density in the soil were found after fungicide treatment. Nonetheless, it is interesting to note that cyazofamid also inhibited *P. brassicae* resting spore germination and reduced spore viability [14].

In conclusion, amisulbrom appeared to be effective at reducing clubroot severity and preserving yields in canola, suggesting that this fungicide could have a role in the integrated management of this disease, likely in conjunction with genetic resistance and other control strategies. Before specific recommendations can be made to growers, however, more research will be required to optimize and validate application methods in broad-acre canola crops.

## 4. Materials and Methods

### 4.1. Amisulbrom 

All products were provided by Gowan Canada (Winnipeg, MB, Canada), including a 20% (*w*/*v*) amisulbrom stock solution, the granular amisulbrom/fertilizer formulations AF700, AF1000, and AF1500 (700, 1000 and 1500 g ai ha^−1^ of amisulbrom, respectively, plus monoammonium phosphate (MAP, 11-52-0 N:P:K)), and fertilizer without amisulbrom. The amisulbrom stock solution with 20% active ingredient was diluted to concentrations from 10% to 0.01% to investigate the effect of the fungicide on *P. brassicae* resting spores and to a 0.1% solution for field and greenhouse applications at 1000 g ai ha^−1^ as the liquid formulation AL1000.

### 4.2. Effect of Amisulbrom on Resting Spore Germination

The effect of amisulbrom on *P. brassicae* resting spore germination was examined using spores suspended in a host root exudate solution. The root exudates were prepared following Macfarlane [35], Suzuki et al. [36], and Lahlali et al. [34] with some modifications. Briefly, 100 seeds of the clubroot-susceptible *B. napus* cv. ‘Westar’ were surface-disinfected with 70% ethanol, rinsed twice in sterile distilled H_2_O, and placed on cheesecloth just immersed near the surface of 100 mL Hoagland’s solution in a 500 mL glass beaker. The top of the beaker was covered with aluminum foil and the beaker was kept at ~25 °C under a 16 h/8 h (light/dark) photoperiod for 14 days. At that time, the cheesecloth and germinated seedlings were discarded, and the solution was adjusted to pH 6.0 and passed through a 0.2 µm syringe filter (VWR International, Mississauga, ON, USA) for use in the spore germination assays. 

Resting spores of a single-spore isolate representing pathotype 3H of *P. brassicae*, as defined on the Canadian Clubroot Differential (CCD) set [7], were extracted from infected roots as described in Strelkov et al. [37]. The spores were then suspended at 5 × 10^7^ spores mL^−1^ in 10 mL of root exudate solution amended with 0, 0.01, 0.1, 1, and 10% amisulbrom in 15 mL Falcon tubes (VWR International, Mississauga, ON). The tubes were incubated in darkness at 25 °C and assessed for germination every 48 h over 10 days. In brief, 25 µL aliquots of the spore suspensions were collected with a micropipette, stained with equal volumes (25 µL) of 2% acetic orcein (Fisher Scientific, Markham, ON) [34,38], and examined on glass slides under a light microscope (Nikon Instruments Inc., Melville, NY, USA). Stained spores were regarded as ungerminated, while nonstained (empty) spores were considered to have germinated. Each treatment was replicated five times (one Falcon tube per replicate), with the percentage germination estimated by evaluating at least 100 spores in three different fields of view [34,38] for each replicate. The experiment was repeated. 

### 4.3. Effect of Amisulbrom on Resting Spore Viability

The effect of amisulbrom on spore viability was assessed in a manner similar to the spore germination assay, except that a suspension of 5 × 10^7^ spores mL^−1^ was generated in 10 mL sterile distilled H_2_O (instead of root exudates) and amended with 0, 0.01, 0.1, 1, and 10% (*w*/*v*) amisulbrom. The resting spores were stained with Evan’s blue [13,39,40] at 2-day intervals over 10 days. Spores with stained cytoplasm were considered dead, while unstained spores were considered viable. This experiment was also repeated.

### 4.4. Field Trials

Field trials were conducted in 2019 (one site) and 2020 (two sites, Site 1 and Site 2) at the Crop Diversification Centre North (53 38′48″ N, 113 22′33″ W), Alberta Agriculture and Irrigation, Edmonton, AB, in dedicated clubroot nurseries. These are biosecure facilities infested with an average 1 × 10^5^ resting spores g^−1^ soil. The three sites were located at a minimum 100 m distance from each other. Treatments were arranged in a split-plot design with four replicates. Two canola hybrids ‘45H31’ (clubroot-susceptible) and ‘CS2000’ (moderately resistant) were seeded on 12 June 2019, and 4 June 2020. The treatments included an untreated control (UTC, fertilizer only), AF700, AF1000, and AF1500, and the liquid formulation AL1000 (Table 5). Fertilizer treatments (with or without amisulbrom) were applied to four 6 m rows per plot at seeding, with approximately 0.7 g of seed planted per row with a push-seeder. The liquid formulation AL1000 was applied to the rows prior to seeding. Since AL1000 did not include any fertilizer, MAP was also included in treatments with this formulation. Untreated control (UTC) treatments received fertilizer alone. Fifteen plants were dug from each plot at eight weeks after seeding and evaluated for clubroot symptom severity as described below. Plant height and aboveground weight were also measured. After plant sampling, soil samples from the top 10 cm layer of each plot were collected for spore density measurement. The trials were harvested on 24 October 2019 and 8 October 2020, and yields were calculated based on grain weight per plot area and recorded.

### 4.5. Greenhouse Trials

Pathogen-free field soil was mixed with Sunshine^®^ Mix #4 Aggregate Plus (Sun Gro, Agawam, MA, USA) at a 1:1 ratio (*v*:*v*) and inoculated with a *P. brassicae* resting spore suspension to produce final spore concentrations of 1 × 10^5^ and 1 × 10^7^ resting spores g^−1^ soil mix. The resting spores were extracted, following Strelkov et al. [37], from clubbed canola roots collected from the nurseries; pathotyping on the CCD set [7] confirmed a pathotype 3H designation. Plastic tubs (43 cm × 28 cm × 17.8 cm) were filled with 4 kg (~8 L) soil mix, and two (30 cm-long) seed rows with 10 cm row spacing were prepared per tub for product application and seeding at a rate of 12 seeds per row. Treatments included the untreated control (UTC), AF700, AF1000, AF1500, and the liquid formulation AL1000 (Table 5). As in the field trials, the canola cultivars ‘45H31’ and ‘CS2000’ were used in the greenhouse tests. The greenhouse was maintained at 20–25 °C (day) and 15–18 °C (night) under natural light supplemented by artificial lighting (16 h light/8 h dark). Two experiments (representing the two different spore concentrations) were set up on different benches in the greenhouse, with four replicates (tubs) of each treatment arranged in a split-plot design. Ten plants were sampled from each tub 8 weeks after seeding and evaluated for disease severity, root gall weight, plant height, and total biomass. The greenhouse trials were independently repeated.

### 4.6. Assessment of Clubroot Severity

Canola roots were rated for clubroot symptom severity on a 0 to 3 scale, where 0 = no galling, 1 = a few galls on lateral roots, 2 = moderate galling on main and lateral roots, and 3 = severe galling on all roots [41]. A disease severity index (DSI) was calculated from the individual plant ratings based on the formula of [42] as modified by Strelkov et al. [37]: $$DSI(\%)=\frac{\sum \;(n\times 0)+(\;n\times 1)+(\;n\times 2)+(\;n\times 3)}{N\times 3}\times 100\%$$
where *n* = number of plants in each rating category and *N* = total number of plants in an experimental unit.

### 4.7. PCR and qPCR Analysis

Soil samples collected from the field plots were air-dried at room temperature and ground to homogeneity in a mortar with a pestle. Total genomic DNA was extracted from 0.25 g of each sample with a DNeasy PowerSoil Kit (Qiagen, Germantown, MD, USA) following the manufacturer’s instructions. The concentration and quality of the DNA samples were determined with a Nanodrop 2000c spectrophotometer (Thermo Fisher Scientific Inc., Waltham, MA, USA). The samples were evaluated for the presence of *P. brassicae* DNA by a conventional PCR analysis following the protocol of Cao et al. [43]. Briefly, 10 ng of DNA was added to a 20 µL PCR reaction with the primers TC1F and TC1R [43]. The amplified products were visualized on a 2% agarose gel. Samples with a band of the expected size (548 bp) were confirmed as positive for *P. brassicae* DNA and subjected to a quantitative PCR (qPCR) analysis according to Rennie et al. [44] with the primers DR1F and DR1R. Briefly, the DNA samples were diluted 10-fold with nuclease-free water and 2.5 µL of the solution was added to a 12.5 µL reaction mixture. Previously quantified *P. brassicae* DNA standards representing five spore densities (1 × 10^2^, 1 × 10^3^, 1 × 10^4^, 1 × 10^5^, and 1 × 10^6^ resting spores g^−1^ soil) were also included in the analysis to relate DNA levels to resting spore densities in soil samples. The reactions were conducted with a StepOnePlus Real Time PCR System (Applied Biosystems, Foster City, CA, USA). 

### 4.8. Data Analysis

Statistical analyses were performed with R 3.6.2 (R Core Team, R Foundation for Statistical Computing, Vienna, Austria). An analysis of variance (ANOVA) was carried out to assess the impact of amisulbrom on resting spore germination or viability and on clubroot development in the field and greenhouse experiments. The Shapiro–Wilk and Bartlett tests were used to validate the normality of the data and homogeneity of the variances, respectively. Data were then compared with Fisher’s LSD test using the ‘Agricolae’ package [45] in R 3.6.2. Differences were considered statistically significant if *p* < 0.05.

## Figures and Tables

**Figure 1 plants-13-00028-f001:**
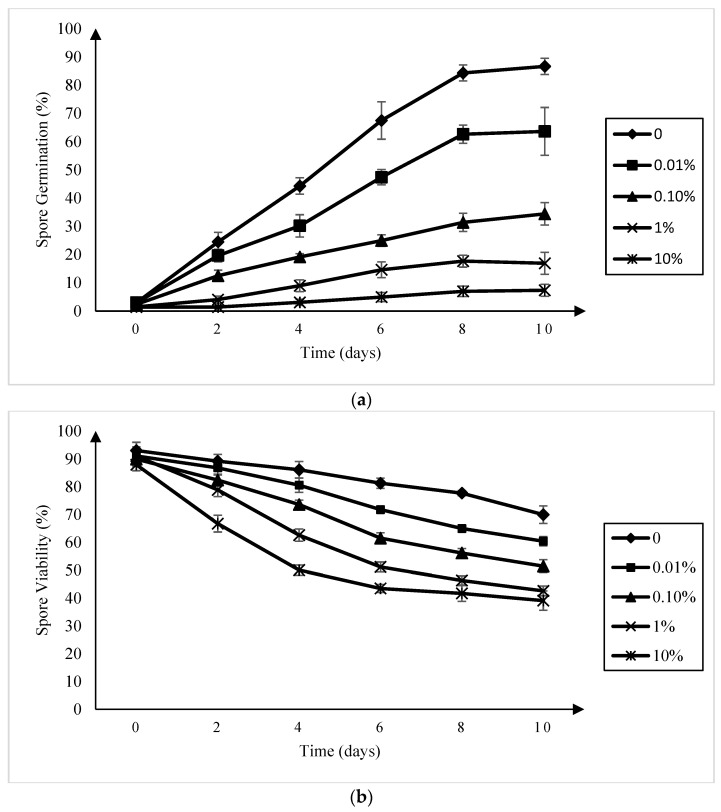
Germination (**a**) and viability (**b**) of *Plasmodiophora brassicae* resting spores in a canola root exudates solution and sterilized water, respectively, amended with amisulbrom at 0, 0.01, 0.1, 1, and 10% (*w*:*v*) over a 10-day period (**b**). Each point indicates the mean ± standard deviation.

**Figure 2 plants-13-00028-f002:**
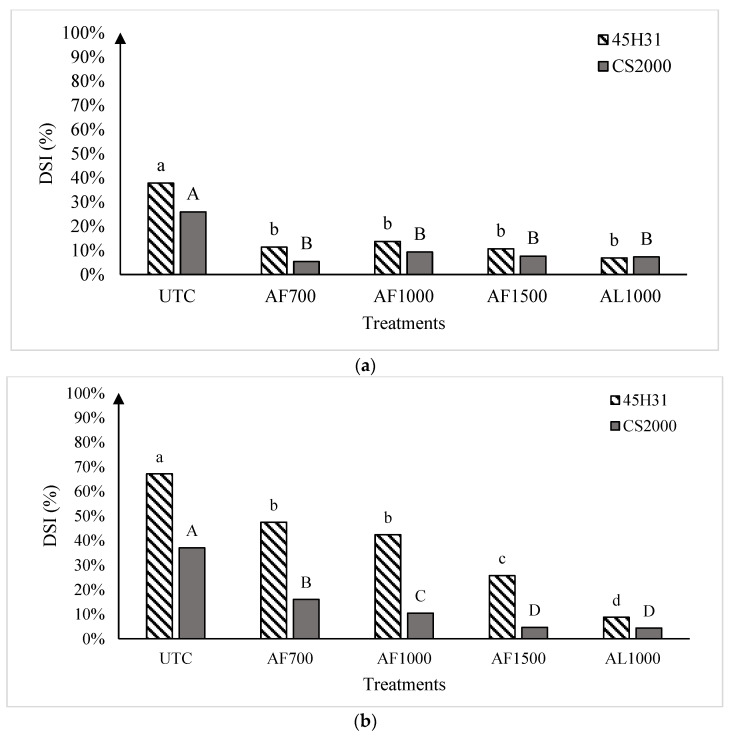
Clubroot disease severity index in the canola hybrids ‘45H31’ and ‘CS2000’ under field conditions in Edmonton in 2019 (panel (**a**)), Edmonton Site 1 in 2020 (panel (**b**)), and Edmonton Site 2 in 2020 (panel (**c**)), following treatment with various amisulbrom rates and formulations. UTC, untreated control; AF700, granular amisulbrom at 700 g active ingredient (ai) ha^−1^; AF1000, granular amisulbrom at 1000 g ai ha^−1^; AF1500, granular amisulbrom at 1500 g ai ha^−1^; AL1000, liquid amisulbrom at 1000 g ai ha^−1^. Bars topped by the same letter are not significantly different at *p* < 0.05; lowercase letters compare differences among treatments for the canola ‘45H31’, while uppercase letters compare differences among treatments for the canola ‘CS2000’.

**Figure 3 plants-13-00028-f003:**
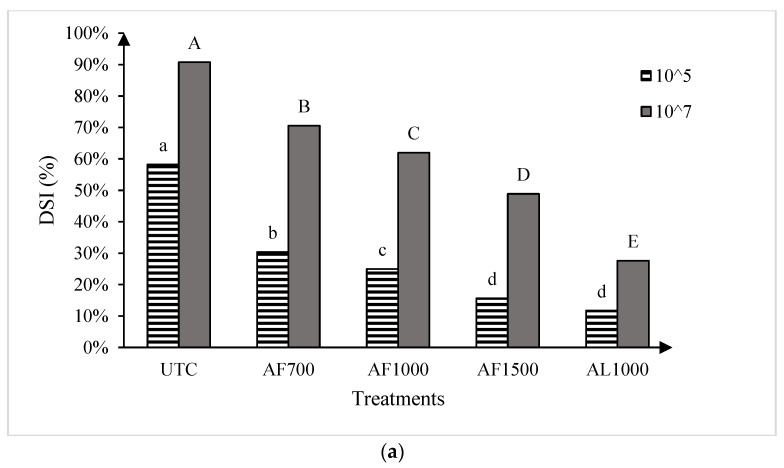
Clubroot disease severity index (DSI) in the canola hybrids ‘45H31’ (**a**) and ‘CS2000’ (**b**) under greenhouse conditions. Treatments were evaluated at two *Plasmodiophora brassicae* resting spore concentrations, 1 × 10^5^ and 1 × 10^7^ resting spores g^−1^ soil mix. UTC, untreated control; AF700, granular amisulbrom at 700 g active ingredient (ai) ha^−1^; AF1000, granular amisulbrom at 1000 g ai ha^−1^; AF1500, granular amisulbrom at 1500 g ai ha^−1^; AL1000, liquid amisulbrom at 1000 g ai ha^−1^. Bars topped by the same letter are not significantly different at *p* < 0.05; lowercase letters compare differences among treatments at 1 × 10^5^ resting spores/g soil mix, while uppercase letters compare differences among treatments at 1 × 10^7^ resting spores/g soil mix.

**Table 1 plants-13-00028-t001:** Average individual plant height, biomass, and yield of the canola hybrid ‘45H31’ in field trials conducted in clubroot-infested soil in Edmonton, Alberta, in 2019 and 2020.

Treatment	Plant Height (cm)			Plant Biomass (g)			Yield (t/ha)		
	2019	2020S1	2020S2	2019	2020S1	2020S2	2019	2020S1	2020S2
UTC	87.81 c	127.48 b	96.68 a	100.89 b	95.65 b	67.23 c	0.95 a	2.48 d	1.10 b
AF700	94.51 b	134.78 ab	99.81 a	110.44 ab	139.13 a	94.36 bc	1.44 a	2.64 cd	1.53 ab
AF1000	94.79 b	133.03 ab	101.81 a	111.71 ab	125.12 ab	85.27 c	1.41 a	2.95 bc	1.55 ab
AF1500	98.47 a	132.91 ab	103.93 a	119.22 a	165.13 a	127.21 ab	1.36 a	3.24 ab	1.88 a
AL1000	96.62 ab	142.82 a	103.15 a	111.17 ab	143.97 a	131.51 a	0.99 a	3.54 a	1.98 a

Notes: UTC, untreated control; AF700, granular amisulbrom at 700 g active ingredient (ai) ha^−1^; AF1000, granular amisulbrom at 1000 g ai ha^−1^; AF1500, granular amisulbrom at 1500 g ai ha^−1^; AL1000, liquid amisulbrom at 1000 g ai ha^−1^; 2020S1, Site 1 in 2020; 2020S2, Site 2 in 2020. Treatments followed by the same letter are not significantly different at *p* < 0.05.

**Table 2 plants-13-00028-t002:** Average individual plant height, biomass, and yield of the canola hybrid ‘CS2000’ in field trials conducted in clubroot-infested soil in Edmonton, Alberta, in 2019 and 2020.

Treatment	Plant Height (cm)			Plant Biomass (g)			Yield (t/ha)		
	2019	2020S1	2020S2	2019	2020S1	2020S2	2019	2020S1	2020S2
UTC	100.08 c	145.10 a	101.47 a	129.84 b	117.88 b	103.75 a	1.51 b	2.83 b	1.48 b
AF700	105.65 b	137.20 a	101.18 a	135.44 b	150.83 ab	114.66 a	2.27 a	3.17 a	1.82 ab
AF1000	105.70 b	140.48 a	100.24 a	142.61 ab	139.56 ab	132.32 a	2.06 ab	3.20 a	1.78 ab
AF1500	107.92 a	144.31 a	105.82 a	157.42 a	175.80 a	148.64 a	1.99 ab	3.35 a	2.23 a
AL1000	105.96 ab	142.33 a	99.91 a	136.19 b	155.83 a	108.75 a	2.18 ab	3.24 a	2.12 ab

Notes: UTC, untreated control; AF700, granular amisulbrom at 700 g active ingredient (ai) ha^−1^; AF1000, granular amisulbrom at 1000 g ai ha^−1^; AF1500, granular amisulbrom at 1500 g ai ha^−1^; AL1000, liquid amisulbrom at 1000 g ai ha^−1^; 2020S1, Site 1 in 2020; 2020S2, Site 2 in 2020. Treatments followed by the same letter are not significantly different at *p* < 0.05.

**Table 3 plants-13-00028-t003:** Average individual plant height and biomass in the canola hybrid ‘45H31’ under greenhouse conditions in a soil mix inoculated with 1 × 10^5^ or 1 × 10^7^ resting spores of *Plasmodiophora brassicae* g^−1^ soil mix.

Treatment	Plant Height (cm)		Plant Biomass (g)	
	10^5^	10^7^	10^5^	10^7^
UTC	83.64 c	74.54 b	14.59 b	9.60 b
AF700	92.94 b	81.68 ab	17.07 b	12.60 ab
AF1000	100.71 a	81.92 ab	23.66 a	11.90 ab
AF1500	101.11 a	82.38 ab	22.30 a	13.73 a
AL1000	86.97 bc	85.47 a	16.01 b	12.85 a

Notes: UTC, untreated control; AF700, granular amisulbrom at 700 g active ingredient (ai) ha^−1^; AF1000, granular amisulbrom at 1000 g ai ha^−1^; AF1500, granular amisulbrom at 1500 g ai ha^−1^; AL1000, liquid amisulbrom at 1000 g ai ha^−1^. Treatments followed by the same letter are not significantly different at *p* < 0.05.

**Table 4 plants-13-00028-t004:** Average individual plant height and biomass in the canola hybrid ‘CS2000’ under greenhouse conditions in a soil mix inoculated with 1 × 10^5^ or 1 × 10^7^ resting spores of *Plasmodiophora brassicae* g^−1^ soil mix.

Treatment	Plant Height (cm)		Plant Biomass (g)	
	10^5^	10^7^	10^5^	10^7^
UTC	89.49 d	79.24 b	15.12 b	12.32 c
AF700	96.32 bc	85.46 ab	21.08 a	14.92 b
AF1000	106.07 a	88.99 a	22.05 a	15.98 ab
AF1500	100.61 b	88.78 a	22.21 a	16.78 a
AL1000	93.15 cd	87.75 a	19.64 a	15.53 ab

Notes: UTC, untreated control; AF700, granular amisulbrom at 700 g active ingredient (ai) ha^−1^; AF1000, granular amisulbrom at 1000 g ai ha^−1^; AF1500, granular amisulbrom at 1500 g ai ha^−1^; AL1000, liquid amisulbrom at 1000 g ai ha^−1^. Treatments followed by the same letter are not significantly different at *p* < 0.05.

**Table 5 plants-13-00028-t005:** List of amisulbrom treatments evaluated in greenhouse and field trials in Edmonton, Alberta, in 2019 and 2020.

Treatment	Amisulbrom Rate	Formulation
UTC	0	N.A.
AF700	700 g ai/ha	Granular
AF1000	1000 g ai/ha	Granular
AF1500	1500 g ai/ha	Granular
AL1000	1000 g ai/ha	Liquid

Notes: AF700, AF1000, and AF1500 refer to 700, 1000, and 1500 g active ingredient (ai) ha^−1^ of a granular formulation of amisulbrom, respectively, which included monoammonium phosphate (MAP; 11-52-0 N:P:K); UTC refers to the untreated control (no amisulbrom, only MAP); AL1000 refers to a liquid formulation of amisulbrom at 1000 g ai/ha, with MAP applied; N.A.: Not Applicable.

## Data Availability

The data presented in the study are included in the article or as Appendix A. Further inquiries can be directed to the corresponding author.

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
