# Peer review of "Evaluation of Amisulbrom Products for the Management of Clubroot of Canola (Brassica napus)"

_plants, 2023, doi:10.3390/plants13010028_

Round 1

Reviewer 1 Report

Comments and Suggestions for Authors

I have had the opportunity to carefully review your paper entitled "Evaluation of Amisulbrom Products for the Management of Clubroot of Canola (Brassica napus)" submitted to the Journal of Plants. Firstly, I would like to commend you for the valuable contribution your research makes to the understanding of clubroot management in canola. Your work addresses a significant issue in agriculture and provides insights that can have practical implications. However, I would like to bring to your attention some points that, in my opinion, need further clarification, discussion, or consideration. I believe addressing these aspects will enhance the overall quality and impact of your work. Please find my comments and queries detailed below.

Reviewer Comments:

In the introduction, it is recommended to highlight the significance of Brassica napus L production and its pivotal role in the economic landscape. Subsequently, attention can be directed towards the challenges that impact these aspects.

Line 23: Rephrase the sentence.

Line 65: to examine the efficacy of granular fertilizer formulation of amisulbrom against clubroot and to compare its effectiveness in a liquid formulation.

Line 304: Did you mean different concentrations ranging from 10% to 0.01%? which formulation is used and how much active ingredient is used in field and greenhouse applications.

Line 322, 370: Strelkov et al. (2006) or Strelkov et al. (2018)? Which one is correct.

Line 338: Were the spores stained with evan’s blue after every 2 days.

Line 87: graph value seems to be different from discussed values in section 2.2.

Line 93-96: rephrase to make it clear.

Line 129: Yield? But you have not mentioned, how to calculate yield in material and method. Root gall weight is not mentioned anywhere in any table, although mentioned in the material and method section.

Line 168: Please mention Figure 3b. Is the reduction in DSI across both cultivars, in comparison with UTC? If YES, then mention it in section 2.4. If NO, how did you make a comparison?

Line 181: Rephrase the sentence.

Line 184: CS2000 plants were 11.1 cm - 16.6 cm taller in the 184 AF1500 and AF1000 treatments (correct statement).

Line 220-224: It was not reflected in the results sections. The material and material section stated that there was one site in 2019 and 2 sites in 2020. You have not mentioned whether it is repeated on the same site in 2020 as in 2019 or different sites. Please make it clear.

Line 222-223: which data showed that the disease severity was reduced?

Line 227-230: any reference?

Line 247-248: Rephrase the sentence so that it becomes meaningful.

Line 286: rephrase

I have checked the references, which are very old, mostly older then 10 years, Reference 19 don’t have a year,

Add the importance of this crop in the start of the introduction, and the very recent citation of 2023 must be cited, such as https://doi.org/10.56946/jspae.v2i1.155 etc

In tables, the statistical letters should not be in superscript

In Table 2, UTC 2019 doesn't have statistical letters 

In Figure 2, the "treatment" should be "Treatments"

Reviewer 2 Report

Comments and Suggestions for Authors

The methodological assumptions and realisation of the objectives of the study are correct. The interpretation of the results is also correct. However, the conclusions should be supplemented.
The amisulbrom (3-(3-bromo-6-fluoro 2-methylindol-1-yl)sulfonyl-N,N-dimethyl-1,2,4-triazole-1-sulfonamide) used in the study is the active ingredient of several fungicides already in use. Among others, it is the active ingredient in one-component formulations such as GENKOTSU or LEIMAY 200 SC and the two-component Zorvec Entecta, which also contains oxathiapiprolin (a compound from the isoxazoline group). Did the granulate and liquid formulation used in the study contain only this compound? If so, please confirm this clearly. It would also be good to complete the information regarding the composition of the fertilisers that were used together with the granulated form. Based on these results, amisulbrom seems promising as a potential fungicide for cabbage clubroot control. Given the importance of Plasmodiophora brassicae in the cultivation of not only oilseed rape but other brassica species, it brings practical information for producers.

Reviewer 3 Report

Comments and Suggestions for Authors

This manuscript describes pot and field trials of formulations of the novel fungicide, amisulbrom, against clubroot disease of canola. Field experiments were conducted over 2 seasons, including 2 sites in the second, more severe, season. Results are consistent across years and sites, and are consistent with pot trials and spore germination trials.

Disease suppression is partial and the authors recommend amisulbrom as a component in integrated disease management packages. Amisulbrom has the potential to reduce primary inoculum in the absence of fumigants such as metam sodium. The authors could point out that reducing primary inoculum is the most effective strategy for monocyclic diseases like clubroot.

I have 2 comments:

1. line 55: I believe amisulbrom is used against downy mildews, but I am not aware of any application to control powdery mildew?

2. Given the discussion of soil pH, did the authors collect any data on soil pH in their trials?
